# The Quantitative Analysis of Water Mass during Winter on the East China Sea Shelf Using an Extended OMP Analysis

**Xiaoshuang Li** [1,2,3,*], **Philip Wallhead** [2], **Richard Garth James Bellerby** [2,3], **Jing Liu** [2,3] and **Anqiang Yang** [3]

1    School of Mathematics and Statistics, Shaanxi Xueqian Normal University, Xi'an 710000, China
2    Norwegian Institute for Water Research, 5006 Bergen, Norway
3    State Key Laboratory of Estuarine and Coastal Research, East China Normal University, Shanghai 200000, China
*    Correspondence: 18717822550@163.com

**Abstract:** The distribution and quantification of water masses on the East China Sea (ECS) shelf is important for identifying and understanding historical climate-driven changes in ocean properties and circulation in the region. We applied an extended Optimum Multiparameter (eOMP) analysis to quantify the relative contribution of water masses using wintertime temperature, salinity, nitrate ($NO_3^-$), phosphate ($PO_4^{3-}$), and silicate ($SiO_3^{2-}$) measurements from a five-cruises dataset spanning from 2013 to 2018. Average ratios ($NO_3^-$:$PO_4^{3-}$:$SiO_3^{2-}$ = 47:1:35) derived from field observations were used to correct the equations referring to the chemical parameters. Our analysis indicated that wintertime seawater on the ECS shelf consisted mainly of Changjiang Dilute Water (CDW), Yellow Sea Coastal Water (YSCW), Taiwan Warm Current Water (TWCW), and East China Sea Shelf Water (ECSSW). The results from the eOMP analysis demonstrated the natural boundaries of four water masses during winter. The interannual variability of water masses showed that the CDW distribution was relatively stable in winter, and there was strong anticorrelation between the YSCW and TWCW extents, suggesting that these two water masses mostly displace each other in the north-south direction.

**Keywords:** water masses; extended Optimum Multiparameter analysis; interannual variability; natural boundary; East China Sea shelf



## 1. Introduction

The East China Sea (ECS) is the largest marginal sea in the western North Pacific Ocean. It is bounded by the Kuroshio in the east, and to the west by continental China from which it receives large amounts of freshwater from the Changjiang (Yangtze) River. It is connected to the South China Sea via the Taiwan Strait, and to the Yellow Sea in the north (Figure 1). The ECS shelf shallower than 200 m covers more than 70% of the entire ECS [1,2]; it is an extremely dynamic shelf region whose circulation system is modulated by the East Asia monsoon [3].

The dominant currents on the ECS shelf present seasonal circulation patterns. It is generally accepted that, in summer, the Changjiang discharge exhibits a bimodal distribution near the river mouth, with a significant fraction of the freshwater flowing to the northeast and the rest flowing southward in a narrow band along the coast [3–5]. In winter, the Changjiang discharge flows southward in a narrow coastal band under the influence of the northerly wind [1,3,4,6]. The Yellow Sea Coastal Current (YSCC), driven by the northerly wind, flows southward along the coast of Jiangsu Province and extends southeastward into the northern part of the ECS [5,7]. The Min-Zhe Coastal Current (MZCC), driven by the winter monsoon, flows southwestward along the ECS coast from Hangzhou Bay to the Taiwan Strait [8]. The Taiwan Warm Current (TWC) is usually defined as the currents originating either in the Taiwan Strait or the Kuroshio intrusion from the northeast of Taiwan. Some studies showed that in summer, the TWC mainly comes from the Taiwan

Strait and flows northeastward parallel to the 50 m isobath and enters the submerged relic river valley off the Changjiang [4,5]. In addition, Zhang et al. [9] and Qi et al. [7] argued that the TWC water results from the mixing of the Taiwan Strait water with the Kuroshio branch northeast of Taiwan in summer. Thus, it is generally accepted that in summer, the surface water of TWC originates from the Taiwan Strait, while the bottom water comes from the Kuroshio Branch Current (KBC) originating from the subsurface water of Kuroshio [10–12]; in winter, the TWC mainly originates from the Kuroshio branch northeast of Taiwan and flows northeastward intruding onto the ECS shelf [13–15]. Over the outer continental shelf, the Kuroshio Current flows northeast along the shelf break [3,4].

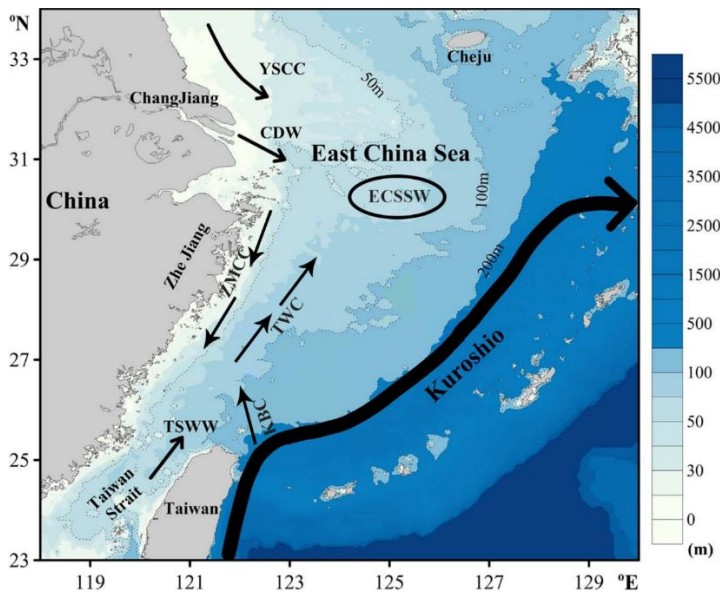

**Figure 1.** The schematic of circulation pattern on the East China Sea (ECS) shelf in winter. Yellow Sea Coastal Current (YSCC), Changjiang Diluted Water (CDW), Taiwan Warm Current (TWC), Zhe-Min Coastal Current (ZMCC), Taiwan Strait Warm Water (TSWW), Kuroshio Branch Current (KBC), and East China Sea Shelf Water (ECSSW).

Numerous studies based on models and observations have reported the spatio-temporal distribution characteristics and mixture of water masses in the ECS [7,8,10,16]. However, these studies described only the qualitative characteristics of water masses on the ECS shelf. Zhang et al. [17] estimated the contribution of individual source water masses to the surface water of the ECS in summer using Optimum Multiparameter (OMP) analysis, and recently the OMP method was used to determine the relative contribution of water masses during summer in the ECS [18]. Although Zhang et al. [10] showed the winter distribution characteristics of water masses in the western ECS shelf area using a cluster analysis method, there remains a lack of common understanding of natural boundaries and the quantitative contribution of water masses during winter, which is of primary importance for understanding biogeochemical systems, such as the pre-bloom nutrient and hydrographic conditions, on the ECS shelf.

This study aims to quantify the mixture contributions, describe natural boundaries, and investigate interannual variability of source water masses on the ECS shelf during winter. The manuscript is organized as follows: The database used in this study, the properties of the main source water masses, and the extended OMP (eOMP) analysis are explained in Section 2. Section 3 contains the mixture contribution of each source water mass, comparison with previous studies and interannual variability of source water masses in winter. The main conclusions are summarized in Section 4.

## 2. Materials and Methods

### 2.1. Biogeochemical Data

Four cruises were conducted during the "Shiptime Sharing Project of National Natural Science Foundation of China", from 4 to 20 March 2013, from 11 to 21 March 2015, from 7 to 19 March 2016, from 15 to 28 February 2017, and one cruise was carried out from 11 to 19 March 2018 during the "National Natural Science Foundation Shared Voyage Plan". Water samples were collected only at the surface in 2013, at the surface and bottom in 2018, and at three different depths during the remaining three cruises. Temperature (T) and salinity (S) profiles were obtained directly using conductivity temperature-depth/pressure (CTD) recorders (SBE 25plus or 911plus). Nutrient samples were first filtered with 0.45 μm Whatman GF/F membrane, then stored in 250 mL HDPE bottles until chemical analysis. Nitrate ($NO_3^-$), phosphate ($PO_4^{3-}$) and silicate ($SiO_3^{2-}$) were determined using a segmented flow analyzer (Model: Skalar SAN$^{plus}$) with a precision <5–10% [9], the detection limits are 0.14 μM for $NO_3^-$, 0.06 μM for $PO_4^{3-}$, and 0.07 μM for $SiO_3^{2-}$, and 762 groups of nutrient data were used. The study area is 122–124° E and 28.5–32.5° N, and the maximum water depth is <100 m. The distribution of the sampling sites is shown in Figure 2.

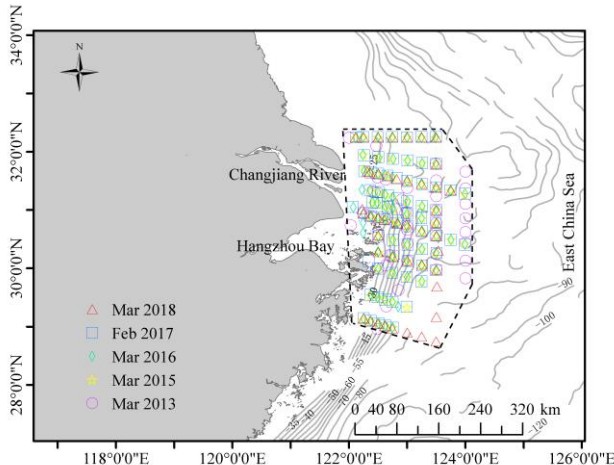

**Figure 2.** Sampling stations during 5 cruises on the ECS shelf. The red triangle, blue square, green diamond, yellow pentagram, and purple circle represent March 2018, February 2017, March 2016, March 2015 and March 2013, respectively. The area inside the dashed black box was chosen to be the study area.

### 2.2. Spatial Distribution and Source Water Type Definition

The distribution features of T and S in the surface and bottom layers on the ECS shelf during winter are shown in Figures 3 and 4, respectively. In general, the distribution characteristics of T and S are similar in the surface and bottom layers: the cold (T < 9 °C) and saline water is located in the northern part of the study area, the fresh (S < 30.5 PSU) water is confined to the western side of the study area, the warm (T > 13.5 °C) and saline water is located the southern part of the study area. This likely reflects strong vertical mixing over the ECS shelf in winter [1,8]. It is worth noting that the bottom water in the southern part was slightly warmer than the surface water during 2015–2017 (Figure 3C vs. Figure 4A, Figure 3D vs. Figure 4B, Figure 3E vs. Figure 4C). In the surface and bottom layers, the cold (T < 9 °C) water with S < 32.5 PSU flows southeastward into the northern part of the ECS shelf, the front of which reaches 31.5–32° N during 2016–2018 (Figures 3D–F,H–J and 4B–D,F–H). This can be identified as the Yellow Sea Coastal Water (YSCW) based on previous observations in winter [5,7,15]. The fresh (S < 30.5 PSU) water outside the Changjiang estuary flows southward and is confined to the coastal area, this is usually identified as the Changjiang Dilute Water (CDW, the mixture of Yangzte River freshwater with saline shelf water). It is interesting to note that in 2015, three stations

((122.27° E, 31.708° N), (122.367° E, 31.651° N), and (122.49° E, 31.623° N)) with the observed surface S greater than 32.5 PSU are located outside the Changjiang estuary (Figure 3G), indicating the upwelling of the bottom water. We can see from Figures 3E,F and 4B–D that an obvious warm (T > 13.5 °C) and saline (S > 33 PSU) water mass, usually identified as the Taiwan Warm Current Water (TWCW), spreads northward along 50 m isobath and reaches ~30.5° N. This feature is consistent with previous results in winter [7,13,15]. Additionally, an apparent saline (S > 32.5 PSU) water mass is located east of 123° E during 2016–2018 (Figures 3H–J and 4F–H), this can be identified as the ECS Shelf Water (ECSSW) described by Ichikawa and Beardsley [1].

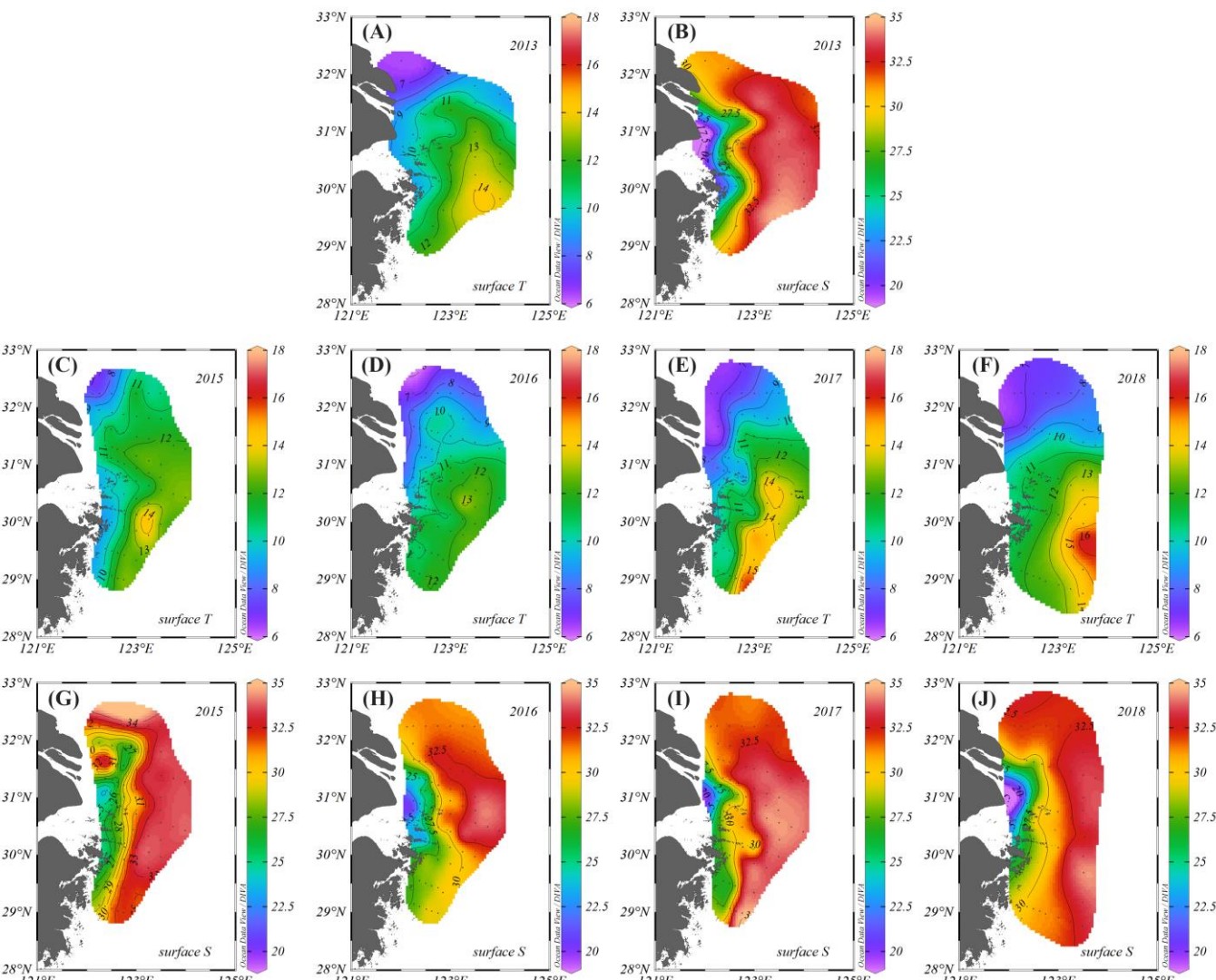

**Figure 3.** Distribution patterns of temperature and salinity in the surface layer (0–3 m) during winter spanning 2013–2018. (**A**) surface T in 2013; (**B**) surface S in 2013; (**C**) surface T in 2015; (**D**) surface T in 2016; (**E**) surface T 2017; (**F**) surface T in 2018; (**G**) surface S in 2015; (**H**) surface S in 2016; (**I**) surface S in 2017; (**J**) surface S in 2018.

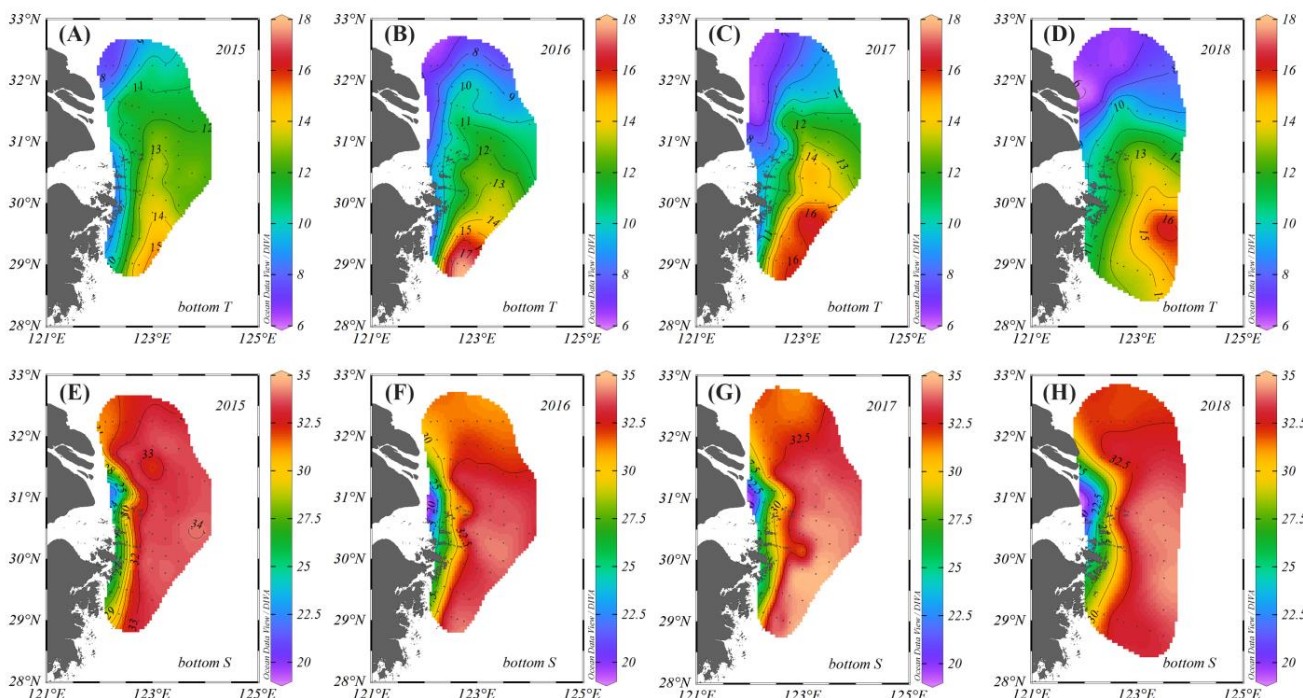

**Figure 4.** Distribution patterns of temperature and salinity in the bottom layer (with water depth ranging from 7 to 75 m following the topography) during winter from 2015 to 2018. (**A**) bottom T in 2015; (**B**) bottom T in 2016; (**C**) bottom T in 2017; (**D**) bottom T in 2018; (**E**) bottom S in 2015; (**F**) bottom S in 2016; (**G**) bottom S in 2017; (**H**) bottom S in 2018.

According to previous research [7,10,13–15,17] and our own observations, the study area is mainly mixed by four prominent water masses: the CDW, the YSCW, the TWCW, and the ECSSW (Figure 5). The CDW has low S (<30.5 PSU) in the western side of the study area, the YSCC is characterized by low T (<9 °C) in the northern part of the ECS shelf, the TWCW is identified as a warm (T > 13.5 °C) water in the southern part of the study area, and the ECSSW is a saline (S > 32.5 PSU) water located east of 123° E.

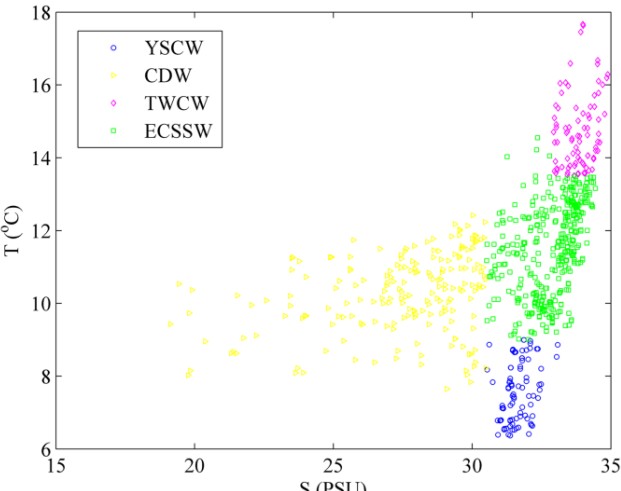

**Figure 5.** The T-S diagram of the data at all stations. The blue circle, yellow triangle, magenta diamond, and green square represent YSCW, CDW, TWCW, and ECSSW, respectively.

### 2.3. Optimum Multiparameter Analysis

OMP analysis, as an extension of the multiparameter analysis proposed by Tomczak [19], was developed to estimate the relative contributions of different water masses by

solving an overdetermined linear system of mixing equations using a non-negative least squares method [20,21]. The physical and chemical properties measured at each point are considered to be the result of the mixing of a certain number of Source Water Types (SWTs), whose physical and chemical characteristics are well-known [22]. OMP analysis includes two main constraints: (a) the mixing contribution of each SWT should be non-negative and the sum of the contributions from all SWTs should be 100% and (b) mass balance should be satisfied at each point. Considering a water mass as a water body with a common element formation history [20], the properties with corresponding standard deviations (STDs) of SWTs were calculated using the scatter of observations in the formation area of the source water masses and are summarized in Table 1. Additionally, the relationships between nitrate, silicate and phosphate in four prominent water masses were fitted with polynomial regressions using in situ observation (Figure 6), the mean ratios ($NO_3^-$:$PO_4^{3-}$:$SiO_3^{2-}$) of nutrient changes in CDW, YSCW, TWCW, and ECSSW were estimated by calculating the gradients at the observed phosphate concentrations, yielding 50:1:36, 20:1:14, 19:1:15, and 39:1:25, respectively.

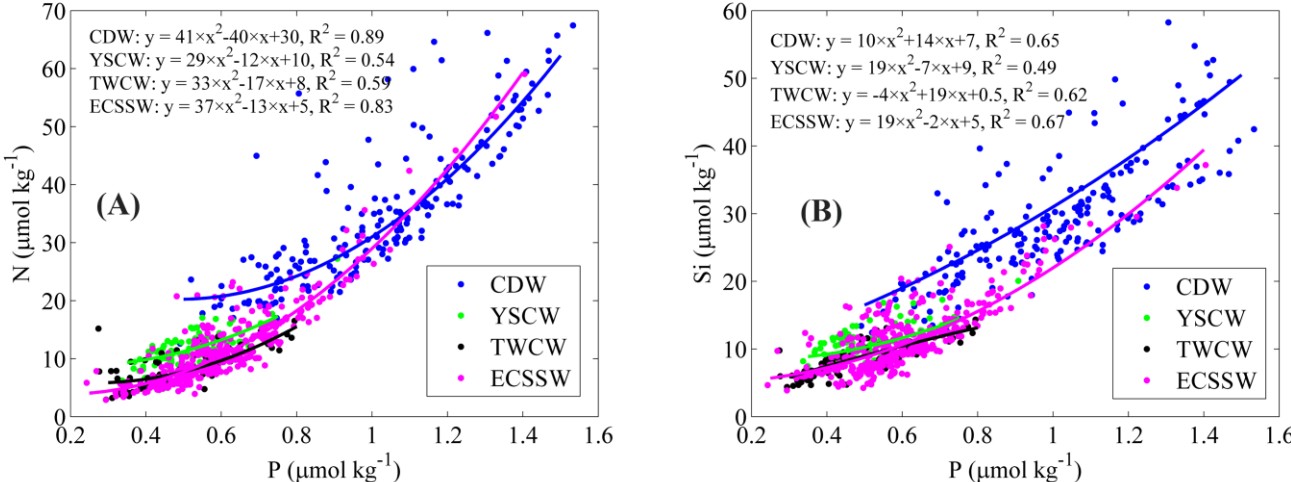

**Figure 6.** The relationships between nitrate (N), silicate (Si) and phosphate (P) in four prominent water masses. The blue, green, yellow, and magenta points represent CDW, YSCW, TWCW, and ECSSW, respectively. (**A**) N-P; (**B**) Si-P.

**Table 1.** Source water type (SWT) characteristics with corresponding standard deviations and regional ratio (mean ± standard error) from the slope (first derivative) values of polynomial regressions in Figure 7. The weights of each equation are also given, together with the square of correlation coefficients ($r^2$) between the observed and estimated properties. Parameters: potential temperature ($\theta$), salinity (S), nitrate ($NO_3^-$), phosphate ($PO_4^{3-}$), and silicate ($SiO_3^{2-}$). CDW, Changjiang Diluted Water; YSCW, Yellow Sea Coastal Water; TWCW, Taiwan Warm Current Water; ECSSW, East China Sea Shelf Water. The last column shows the mean uncertainties in the derived SWT contributions resulting from the uncertainties in SWT characteristics.

| SWT | $\theta$ (°C) | S (PSU) | $NO_3^-$ (µmol kg$^{-1}$) | $PO_4^{3-}$ (µmol kg$^{-1}$) | $SiO_3^{2-}$ (µmol kg$^{-1}$) | Uncertainty |
|---|---|---|---|---|---|---|
| CDW | 8.7 (±0.8) | 18.8 (±1.2) | 60 (±9) | 1.4 (±0.15) | 52 (±7) | 0.01 |
| YSCW | 5.5 (±0.5) | 31.2 (±0.3) | 9.0 (±0.9) | 0.35 (±0.04) | 8.0 (±0.9) | 0.04 |
| TWCW | 17.2 (±0.5) | 34.6 (±0.2) | 7.4 (±0.4) | 0.52 (±0.07) | 5.5 (±0.4) | 0.06 |
| ECSSW | 11.5 (±0.5) | 33.7 (±0.3) | 3.6 (±0.3) | 0.25 (±0.04) | 4.7 (±0.3) | 0.05 |
| Weight | 38.5 | 37.3 | 8.8 | 12.3 | 10.8 | |
| Ratio | N/A | N/A | 47 (±7) | 1 | 35 (±4) | |
| $r^2$ | 0.99 | 0.99 | 0.95 | 0.96 | 0.95 | |

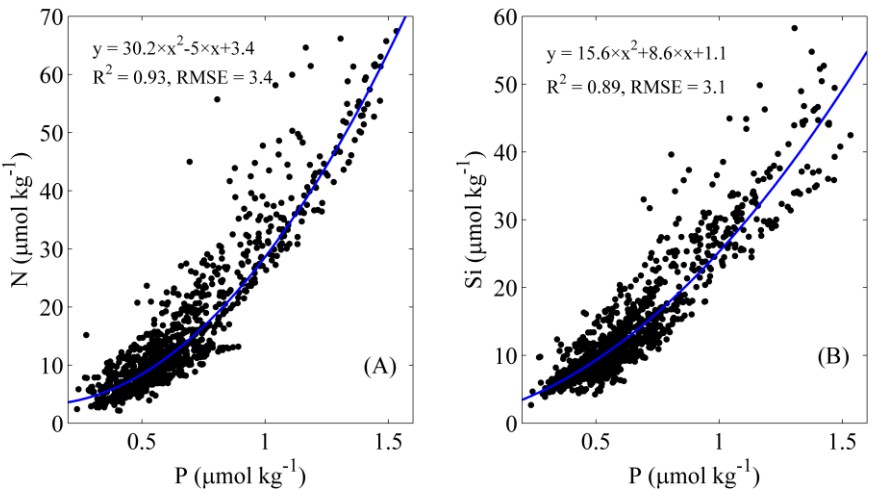

**Figure 7.** The relationships between nitrate (N), silicate (Si) and phosphate (P) from all observations. (**A**) N-P; (**B**) Si-P.

An extended OMP analysis (eOMP) was applied here using both conservative (potential temperature-θ and S) and non-conservative ($NO_3^-$, $PO_4^{3-}$, and $SiO_3^{2-}$) variables by adding a biogeochemical term (ΔP in this work) related to the non-conservative variable in order to account for the biogeochemical processes, through a predefined stoichiometric coefficient ($r_{N/P}$, $r_{Si/P}$). The system of equations is

$$\sum_{i=1}^{n} X_i * \theta_i^{SWT} = \theta^{obs} + R_\theta \tag{1}$$

$$\sum_{i=1}^{n} X_i * S_i^{SWT} = S^{obs} + R_S \tag{2}$$

$$\sum_{i=1}^{n} X_i * [NO_3]_i^{SWT} + r_{N/P}\Delta P = [NO_3]^{obs} + R_{[NO_3]} \tag{3}$$

$$\sum_{i=1}^{n} X_i * [PO_4]_i^{SWT} + \Delta P = [PO_4]^{obs} + R_{[PO_4]} \tag{4}$$

$$\sum_{i=1}^{n} X_i * [SiO_3]_i^{SWT} + r_{Si/P}\Delta P = [SiO_3]^{obs} + R_{[SiO_3]} \tag{5}$$

$$\sum_{i=1}^{n} X_i = 1 + R_{mass} \tag{6}$$

where $R_p$ is the fit residuals of each property $p$ (θ, S, $NO_3^-$, $PO_4^{3-}$, and $SiO_3^{2-}$), $p^{obs}$ is observational data, $p_i^{SWT}$ is the property of each SWT, and $X_i$ is the contribution of the source water type. Equation (6) accounts for mass conservation. The relationships between nitrate, silicate and phosphate were fitted with polynomial regressions using in situ observation (Figure 7). Mathematically the ratio can be expressed by the first derivative of fitted polynomial, the mean value and corresponding standard errors (SE) of the slope ($r_{N/P}$, $r_{Si/P}$) were estimated by calculating the gradients at the observed phosphate concentrations (Table 1). Due to the differences in measurement accuracy and environmental variability of each parameter, the weight for each variable was calculated using the approach described by Tomczak and Large [20], which is the ratio of the parameter's variance in the SWT matrix and the maximum variance of the parameter considering all water masses. The weights used are summarized in Table 1. The SWT matrix and Redfield ratios were normalized to make the units of different parameters commensurable. The code is available from http://omp.geomar.de, accessed on 10 October 2022. The OMP method has been applied

to evaluate the mixing contributions of water masses on regional [17,18,23–25], ocean basin [21,22,26], and global [27] scales.

### 2.4. Uncertainty

The mass conservation residuals from fitting the five cruises of data are shown in Figure 8F. All residuals are lower than 7%, 89% of the residuals are lower than 1%, and only four residuals are higher than 3%. The squared correlation coefficients between observed and estimated properties by eOMP analysis using average ratios ($NO_3^-:PO_4^{3-}:SiO_3^{2-}$ = 47:1:35) are shown in Figure 8A–E. The model was considered reliable since it explains almost 95% of the variability of the conservative and non-conservative tracers.

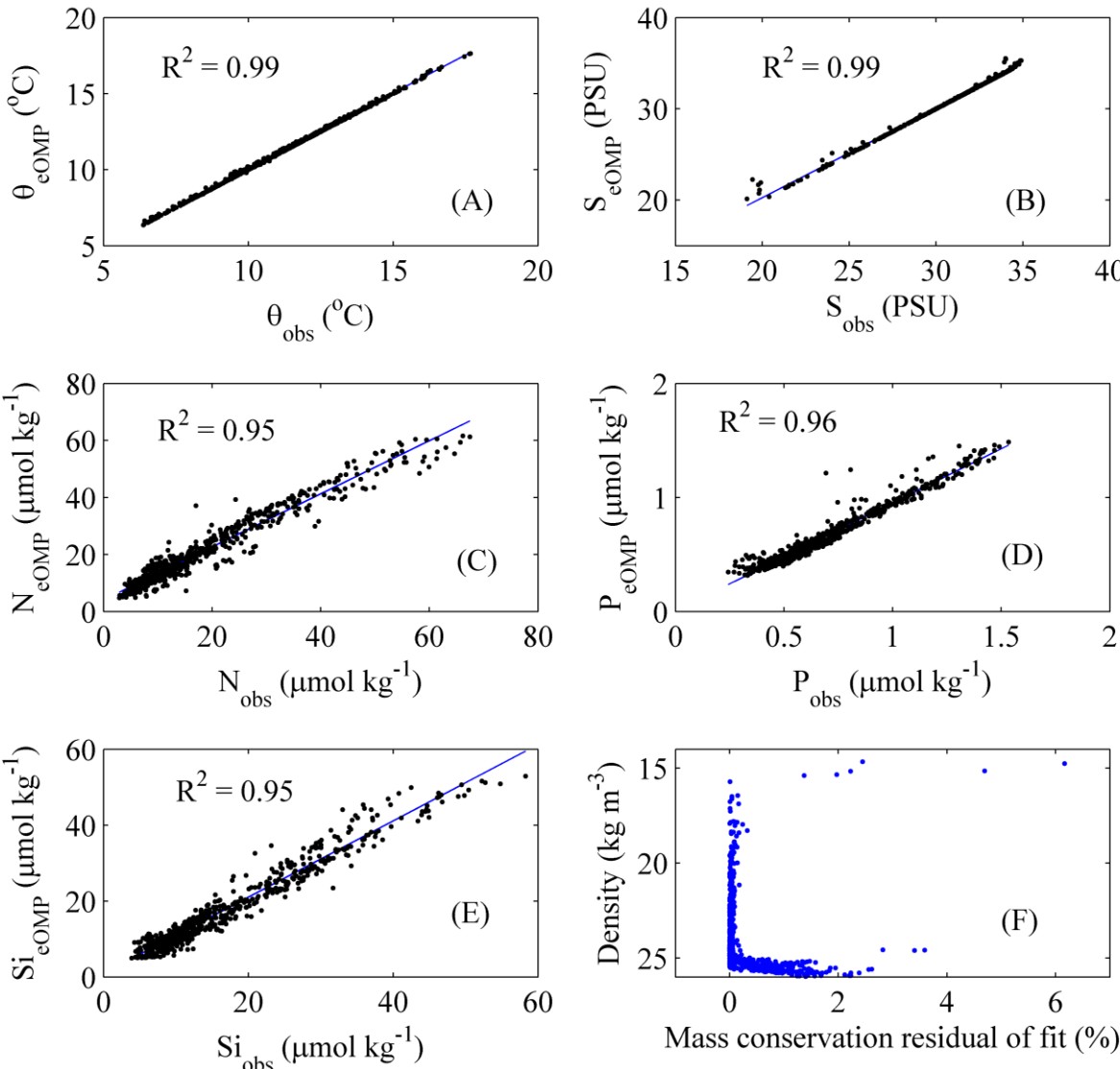

**Figure 8.** The square of correlation coefficient between the observed (subscript obs) and estimated (subscript eOMP) properties by eOMP analysis: (**A**) potential temperature-θ; (**B**) salinity-S; (**C**) nitrate-N; (**D**) phosphate-P; (**E**) silicate-Si. (**F**) The mass conservation residuals computed by eOMP analysis versus density.

The robustness of the eOMP analysis was tested through a perturbation analysis of uncertainties using a Monte Carlo analysis. Here the properties of each SWT were randomly perturbed using normal distributions with the standard deviations shown in Table 1. In total, 100 perturbations were performed and the eOMP analysis was solved

for each perturbed system. We calculated the STD over the 100 eOMP runs performed. Figure 9A shows the distribution of uncertainties in the contributions with depth for the CDW, YSCW, TWCW, ECSSW, and the biogeochemical term ΔP. The average values of the uncertainties of $X_{CDW}$, $X_{YSCW}$, $X_{TWCW}$ and $X_{ECSSW}$ were 0.01 (1%), 0.04 (4%), 0.06 (6%) and 0.05 (5%) respectively (Table 1), suggesting that the methodology is fairly robust.

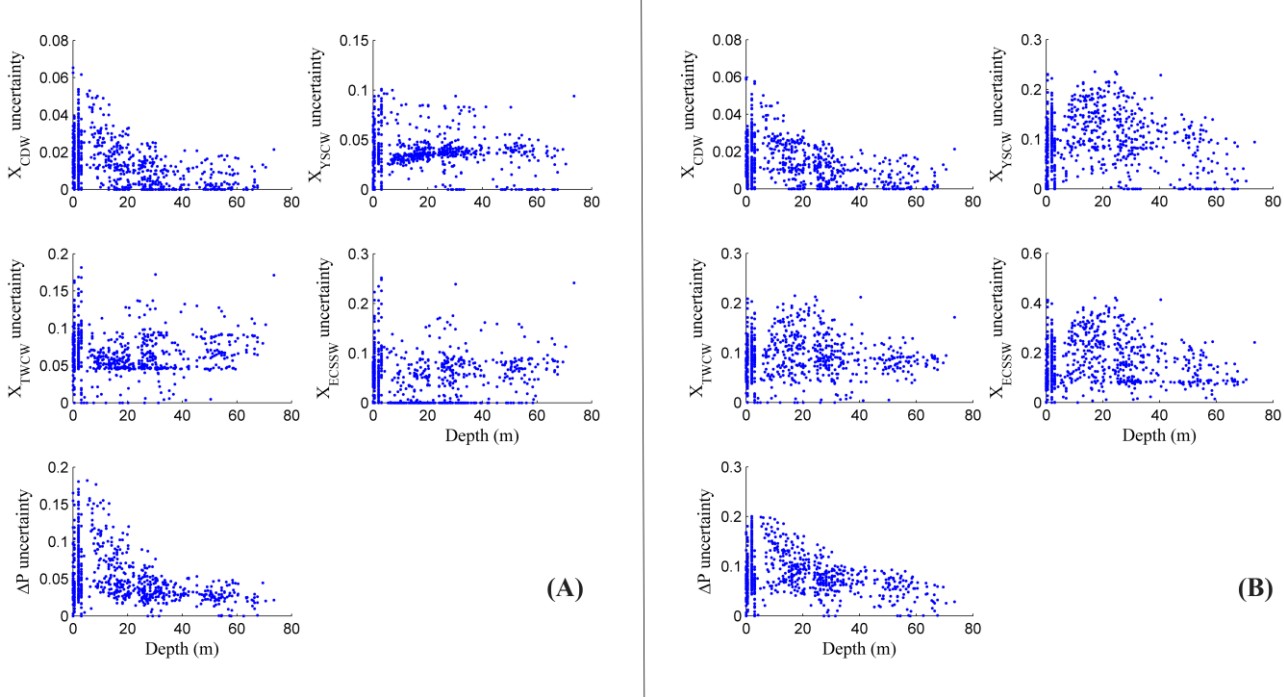

**Figure 9.** The uncertainties (one ensemble standard deviation) of the contributions (X) with depth for the CDW, YSCW, TWCW, ECSSW, and the biogeochemical term ΔP. (**A**) derived from a perturbation analysis of SWT properties; (**B**) derived from a perturbation analysis of SWT properties and stoichiometric coefficient.

We also tested the sensitivity to the combined uncertainty in SWT properties and Redfield ratios ($NO_3^-$:$PO_4^{3-}$:$SiO_3^{2-}$). First, we obtained 100 perturbations for each SWT using a Monte Carlo analysis; then we chose one ratio among the four SWT-specific ratios (50:1:36, 20:1:14, 19:1:15, and 39:1:25) by a random choice for every perturbation. Each perturbed system was solved, and the STD of each ensemble member was calculated. The distribution of uncertainties in the contributions is shown in Figure 9B. The average values of the uncertainties of $X_{CDW}$, $X_{YSCW}$, $X_{TWCW}$ and $X_{ECSSW}$ were 0.01 (1%), 0.10 (10%), 0.09 (9%) and 0.16 (16%) respectively. Compared to Figure 9A, the uncertainty of the CDW remains the same, while the uncertainties of the YSCW and ECSSW have increased more than one time and two times, respectively. This suggests that the uncertainty in the Redfield ratios was a significant source of uncertainty in the SWT contributions except for the CDW.

## 3. Results and Discussion

### 3.1. Contributions and Natural Boundaries of Source Water Masses

To discriminate natural boundaries of source water masses, the area where any SWT contribution exceeds 50% was assigned to that SWT, and the area where all SWT contributions were <50% was considered to be the Mixed Water (MW) area, as described by Zhou et al. [18]. The mixture contributions of the source water masses on the ECS shelf during winter are shown in Figures 10 and 11. The CDW was confined to the west of 122.5° E and formed a narrow belt along the coast, and its contribution gradually decreased with distance offshore. The highest contribution of the CDW was 97% in March 2013 (Figure 10A),

70% in March 2015 (Figures 10E and 11A), 89% in March 2016 (Figures 10I and 11E), 85% in February 2017 (Figures 10M and 11I), and 92% in March 2018 (Figures 10Q and 11M). The contribution of the CDW decreased to <30% inside the 50 m isobath and the distribution showed limited interannual variability, suggesting that the wintertime CDW distribution was stable during 2013–2018.

The YSCW usually dominated north of 31.5° N. The highest contribution was 88% in March 2013 (Figure 10B), 82% in March 2015 (Figures 10F and 11B), 86% in March 2016 (Figures 10J and 11F), 91% in February 2017 (Figures 10N and 11J), and 84% in March 2018 (Figures 10R and 11N). Close to 31.5° N, the contribution decreased to ~30% in March 2013, ~40% in March 2015 and 2018, and ~50% in March 2016 and February 2017. Additionally, the area with YSCW contribution <30% reached down to ~30.5° N in March 2016 and February 2017. This indicates that the YSCW was stronger in March 2016 and February 2017 than in March 2013, 2015, and 2018.

The TWCW usually dominated south of 30.5° N, and the surface contribution was obviously weaker than the bottom contribution during 2015–2017. For example, the highest contribution of the surface TWCW was 59% in March 2016 (Figure 10K) and 60% in February 2017 (Figure 10O), but the highest contribution of the bottom TWCW was ~100% in March 2016 (Figure 11G) and 83% in February 2017 (Figure 11K).

The ECSSW was mainly located east of 123° E, and the surface contribution was weaker than the bottom contribution during 2015–2018. For example, the highest contribution of the surface ECSSW was ~95% in February 2017 (Figure 10P) and ~73% in March 2018 (Figure 10T), but the highest contribution of the bottom ECSSW was ~97% in February 2017 (Figure 11L) and ~98% in March 2018 (Figure 11P); the coverage area of the bottom ECSSW was greater than the surface ECSSW in March 2015 (Figure 10H vs. Figure 11D) and 2016 (Figure 10L vs. Figure 11H). In March 2013, the coverage area where the surface ECSSW contributed >50% was the largest, accounting for ~68% of the study area (Figure 10D).

### 3.2. Comparison with Previous Studies

Previous qualitative studies of winter water masses distribution on the ECS shelf [7,10,15] can be used as a point of reference for our eOMP results. These have suggested that both T and S are vertically well-mixed without obvious stratification [1,4,10,15]. However, the bottom T was obviously warmer than the surface T in the southern part of the study area during 2015–2017 (Figure 3C vs. Figure 4A, Figure 3D vs. Figure 4B, Figure 3E vs. Figure 4C), indicating a greater bottom invasion of warm, saline water. This was clearly reflected in our eOMP results: the bottom TWCW contribution was obviously stronger than the surface contribution in March 2016 and February 2017 (Figure 10G vs. Figure 11C, Figure 10K vs. Figure 11G, Figure 10O vs. Figure 11K).

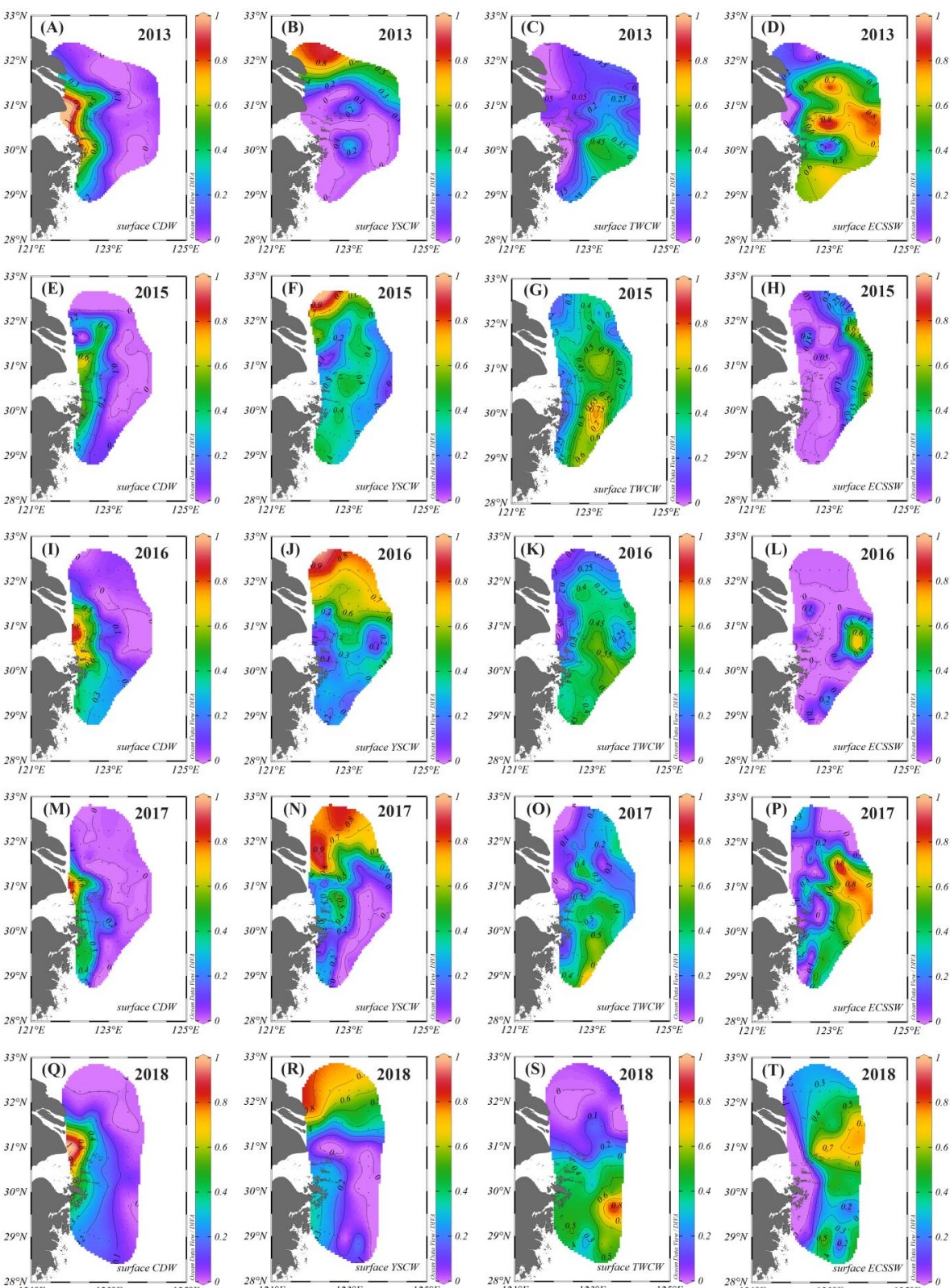

**Figure 10.** The mixture contribution of the source water masses in the surface layer of the ECS shelf spanning from 2013 to 2018. (**A**) CDW in 2013; (**B**) YSCW in 2013; (**C**) TWCW in 2013; (**D**) ECSSW in 2013; (**E**) CDW in 2015; (**F**) YSCW in 2015; (**G**) TWCW in 2015; (**H**) ECSSW in 2015; (**I**) CDW in 2016; (**J**) YSCW in 2016; (**K**) TWCW in 2016; (**L**) ECSSW in 2016; (**M**) CDW in 2017; (**N**) YSCW in 2017; (**O**) TWCW in 2017; (**P**) ECSSW in 2017; (**Q**) CDW in 2018; (**R**) YSCW in 2018; (**S**) TWCW in 2018; (**T**) ECSSW in 2018.

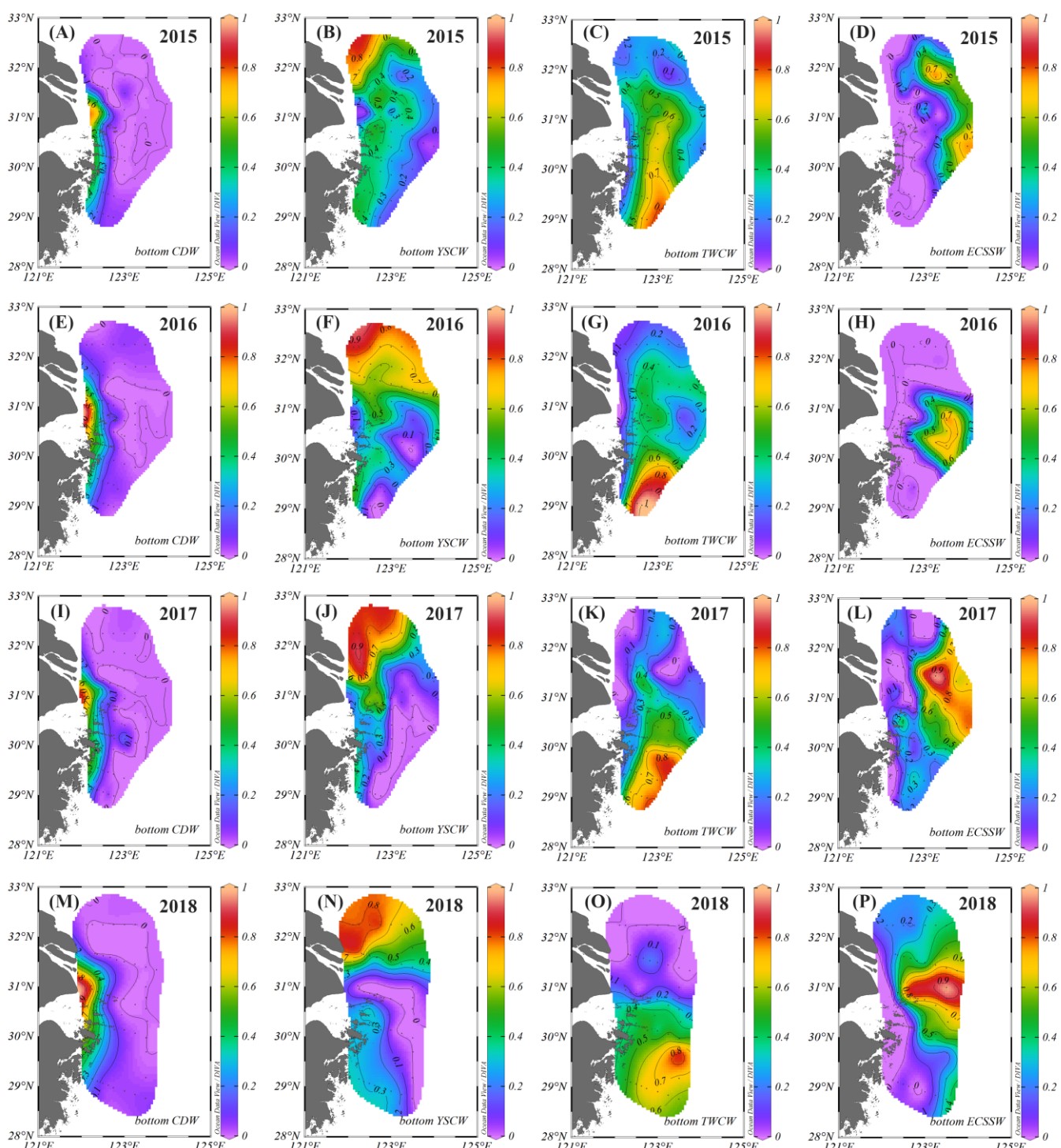

**Figure 11.** The mixture contribution of the source water masses in the bottom layer of the ECS shelf from 2015 to 2018. (**A**) CDW in 2015; (**B**) YSCW in 2015; (**C**) TWCW in 2015; (**D**) ECSSW in 2015; (**E**) CDW in 2016; (**F**) YSCW in 2016; (**G**) TWCW in 2016; (**H**) ECSSW in 2016; (**I**) CDW in 2017; (**J**) YSCW in 2017; (**K**) TWCW in 2017; (**L**) ECSSW in 2017; (**M**) CDW in 2018; (**N**) YSCW in 2018; (**O**) TWCW in 2018; (**P**) ECSSW in 2018.

Our analysis suggested that the CDW was confined to west of 122.5° E, which is consistent with previous wintertime observations. Zhang et al. [10] found that the CDW was distributed in the offshore area west of 122.8° E in 2009, and Lian et al. [15] argued that the CDW was confined to the coastal area west of 122.5° E in 2014. Our analysis also

suggested that the YSCW was dominant north of 31.5° N. This is consistent with previous results for winter: Qi et al. [7] showed that the YSCW flowed southward into the ECS to near 32° N in 2011, while Lian et al. [15] showed that the YSCC was restricted to the shallow regions (north of 31.75° N) along the coast in 2014. Our analysis showed that the TWCW mainly dominated south of 30.5° N and flowed northeastward, parallel to the 50 m isobath intruding onto the ECS shelf. This feature is also supported by previous wintertime observations. Using both hydrographic and current measurements from three anchored ADCPs, Zhu et al. [13] clearly showed that the TWCW spread northeastward along the 50 m isobath and intruded into the submerged river valley off the Changjiang Estuary in 2001, reaching as far as 30.5° N in 2014 [15], and up to ~31° N in 2011 [7]. Although it may be questionable that the YSCW may contribute >30% south of 30.5° N (Figures 10F,J,N and 11B,F,J,N) and that the TWCW may contribute >30% north of 31.5° N (Figures 10G,K,O and 11C,G,K), such results are not implausible as source water masses can spread far from their formation regions before significant transformation occurs [26]. We considered four water masses during winter on the ECS shelf, while Zhang et al. [10] only considered three, omitting the ECSSW. In our eOMP results, the ECSSW was mainly located east of 123° E, and the coverage area of the bottom ECSSW was greater than the surface ECSSW in March 2015 (Figure 10H vs. Figure 11D) and 2016 (Figure 10L vs. Figure 11H).

### 3.3. Interannual Variability of Source Water Masses

To quantify the interannual variability of source water masses, we calculated the percentage of SWT area at the surface and bottom in the study region from 2015 to 2018 (Table 2). The percentage of the CDW area was less than 10% except for at the surface in 2016, and the standard deviation of the CDW over the 4 years was less than 4%, suggesting that the CDW distribution was relatively stable in winter. The percentage of the ECSSW area in 2015 and 2016 was smaller than in the next two years. The percentage of the YSCW area in 2015 was smaller than in the next two years, while the percentage of the TWCW area in 2015 was greater than in the next two years. Despite the limited number of years considered ($n = 4$), there are two strong anticorrelations (linear, Pearson) between YSCW and TWCW at the surface (r = $-0.96$, $p = 0.035$) and bottom (r = $-0.98$, $p = 0.024$), suggesting that these two water masses mostly displace each other in the north-south direction. The source water mass with the greatest standard deviation over the 4 years was the surface ECSSW, followed by the YSCW and TWCW.

**Table 2.** Percentage of source water type (SWT) area in study region from 2015 to 2018. CDW, Changjiang Diluted Water; YSCW, Yellow Sea Coastal Water; TWCW, Taiwan Warm Current Water; ECSSW, East China Sea Shelf Water. The last row lists the standard deviation (STD) over the 4 years. Note the sum of these percentages was far from 100% because the area where all SWT contributions were <50% was considered to be the Mixed Water area.

| Year | CDW | | YSCW | | TWCW | | ECSSW | |
|------|---------|--------|---------|--------|---------|--------|---------|--------|
|      | Surface | Bottom | Surface | Bottom | Surface | Bottom | Surface | Bottom |
| 2015 | 7.7%    | 4.3%   | 6.2%    | 11.4%  | 26.2%   | 37.1%  | 3.1%    | 12.9%  |
| 2016 | 15.1%   | 4.2%   | 35.6%   | 38.9%  | 8.2%    | 15.3%  | 2.7%    | 12.5%  |
| 2017 | 5.6%    | 2.9%   | 29.6%   | 30.4%  | 7.0%    | 21.7%  | 29.6%   | 27.5%  |
| 2018 | 6.8%    | 6.7%   | 18.2%   | 22.2%  | 15.9%   | 24.4%  | 36.4%   | 35.6%  |
| STD  | 3.7%    | 1.4%   | 11.2%   | 10.2%  | 7.7%    | 7.9%   | 15.2%   | 9.9%   |

### 4. Conclusions

This study represents a regional application of the extended Optimum Multiparameter (eOMP) analysis to quantify the mixture contribution of water masses in winter on the ECS shelf, based on an extensive dataset collected during five cruises between 2013 and 2018. To further constrain our analysis, we used average ratios ($NO_3^-:PO_4^{3-}:SiO_3^{2-}$ = 47:1:35) derived from field observations on the ECS shelf to correct the equations for

variability in fluvial nutrient supply and stoichiometry of biological activity. The mass conservation residuals and uncertainties in the four water mass fractions from Monte Carlo analysis were low (<10% on average over the region). Our analysis demonstrated the natural boundaries of water masses during winter: the CDW was confined to the west of 122.5° E, the YSCW dominated north of 31.5° N, the TWCW dominated south of 30.5° N, and the ECSSW was mainly located east of 123° E. Overall, the results were consistent with previous studies, but provided a more detailed and quantitative view of the water mass distributions on the ECS shelf in winter. The CDW showed limited wintertime interannual variability. There was a strong interannual anticorrelation between the areal extents of YSCW and TWCW, suggesting that these two water masses mostly displace each other in the north-south direction. Future work should try and account for potential forcings on this variability including changes in local, regional and large-scale circulation, wind fields, and fluvial contributions. The eOMP analysis appeared to be a powerful tool for estimating the contribution and mixing of water masses. Our results may also be relevant to monitoring and understanding climate-driven changes in circulation on the ECS shelf, which is considered to be a region of high sensitivity to anthropogenic influences.

**Author Contributions:** Conceptualization, X.L., P.W. and R.G.J.B.; methodology, X.L., P.W. and R.G.J.B.; software, X.L.; validation, X.L.; formal analysis, X.L., P.W. and R.G.J.B.; investigation, X.L., J.L. and A.Y.; resources, X.L., J.L. and A.Y.; data curation, X.L., J.L. and A.Y.; writing—original draft preparation, X.L., P.W. and R.G.J.B.; writing—review and editing, X.L., P.W. and R.G.J.B.; visualization, X.L.; supervision, P.W. and R.G.J.B.; project administration, X.L., P.W. and R.G.J.B.; funding acquisition, X.L., P.W. and R.G.J.B. All authors have read and agreed to the published version of the manuscript.

**Funding:** This research was funded by the National Thousand Talents Program for Foreign Experts (grants No. WQ20133100150), Vulnerabilities and Opportunities of the Coastal Ocean (grants No. SKLEC-2016RCDW01), Marginal Seas (MARSEAS) (grants SKLEC-Taskteam project), Innovative Talents International Cooperation Training Project (grants No. China Scholarship Council-201913045), Introducing Talents (Ph.D.) Research Projects (grants No. 2021DS05), and Scientific Research Program Funded by Shaanxi Provincial Education Department (Program No. 22JK0335). R.G.J.B. and P.W. were also supported by the FRAM High North Research Centre for Climate and the Environment under the Research Program CLEAN and the NIVA Land-Ocean Interactions Strategic Institute program.

**Data Availability Statement:** Matlab code of the eOMP analysis is available from http://omp.geomar.de, accessed on 10 October 2022. Requests to access the raw data should be directed to Jianzhong Ge: jzge@sklec.ecnu.edu.cn. The results of the eOMP analysis, perturbation analysis of SWT properties, and perturbation analysis of SWT properties and stoichiometric coefficient are available: http://doi.org/10.5281/zenodo.3934943, accessed on 10 October 2022.

**Acknowledgments:** We are grateful to Jianzhong Ge for assistance with providing four cruises data on the East China Sea shelf from 2013 to 2017. We sincerely thank the people who worked on the cruises and in the laboratory.

**Conflicts of Interest:** The authors declare no conflict of interest.

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
