# Peer review of "The Quantitative Analysis of Water Mass during Winter on the East China Sea Shelf Using an Extended OMP Analysis"

_water, doi:10.3390/w14203214_

Round 1
Reviewer 1 Report
The present manuscript applies the extended Optimum Multiparameter Analysis to the analysis of the distribution of the main water masses in a continental shelf region of East China Sea. This oceanographic method of data analysis is not new, but it is useful to provide reference studies describing the biogeochemical characteristics of source water types and of their displacement. It is a positive aspect that this study also consider the interannual variability of this coastal system.
The manuscript is written properly and it is suitable for the publication on WATER. I have only few minor suggestions for its revision:
Text and captions would be clearer matching always the acronyms and their definitions. For example, in the caption of Figure 1:
Figure 1. Schematic of circulation pattern on the East China Sea (ECS) shelf in winter: Yellow Sea Coastal Current (YSCC), Changjiang Diluted Water (CDW), Taiwan Warm Current (TWC), Zhe-Min Coastal Current (ZMCC), Taiwan Strait Warm Water (TSWW), Kuroshio Branch Current (KBC) and East China Sea Shelf Water (ECSSW).
In the section “2.1. Biogeochemical Data”, the authors should specify the number of CTD downcasts and nutrient data used in this study.
Table 1: Please, explain better in the Table 1 or in its caption what are the data shown in the row “ratio”. They should be means and standard errors of the slope (first derivative) values of the fitting curves of NO3/PO4, PO4/PO4, SiO3/PO4 plots shown in Figure 7.
Reviewer 2 Report
The authors have submitted an interesting manuscript that reports the quantitative analysis of water mass during winter on the East China Sea shelf using an extended OMP analysis. In my opinion is an outstanding and complete case study that considering the aim, methodology and results merit to be accepted for publication in WATER without additional corrections in its present form.
Author Response
Thank you for your approval and review of our manuscript.